# Measurement of Perfusion Heterogeneity within Tumor Habitats on Magnetic Resonance Imaging and Its Association with Prognosis in Breast Cancer Patients

**DOI:** 10.3390/cancers14081858

**Published:** 2022-04-07

**Authors:** Hwan-ho Cho, Haejung Kim, Sang Yu Nam, Jeong Eon Lee, Boo-Kyung Han, Eun Young Ko, Ji Soo Choi, Hyunjin Park, Eun Sook Ko

**Affiliations:** 1Department of Medical Artificial Intelligence, Konyang University, Daejon 32992, Korea; nara9313@gmail.com; 2Center for Neuroscience Imaging Research, Institute for Basic Science (IBS), Sungkyunkwan University, Suwon 16419, Korea; 3Department of Radiology, Samsung Medical Center, Sungkyunkwan University School of Medicine, Seoul 06351, Korea; hjk220@naver.com (H.K.); bkhan@skku.edu (B.-K.H.); claudel@skku.edu (E.Y.K.); jisoo.choi@samsung.com (J.S.C.); 4Department of Radiology, Gil Hospital, Gachon University of Medicine and Science, Incheon 21565, Korea; sangyu.nam7@gmail.com; 5Department of Surgery, Samsung Medical Center, Sungkyunkwan University School of Medicine, Seoul 06351, Korea; jeongeon.lee@samsung.com; 6School of Electronic and Electrical Engineering, Sungkyunkwan University, Suwon 16419, Korea

**Keywords:** breast cancer, magnetic resonance imaging, perfusion, heterogeneity, kinetics, radiomics, prognosis

## Abstract

**Simple Summary:**

A habitat analysis reflects intratumoral heterogeneity more accurately than does a whole-tumor analysis. Perfusional heterogeneity using a habitat analysis is a rarely explored option and can affect patient outcomes. From two hospitals, 308 and 147 patients with invasive breast cancer who underwent preoperative MRI were included as development and validation cohorts, respectively. In our study, five habitats with distinct perfusion patterns were identified based on early and delayed phases of dynamic contrast material-enhanced MR images. A habitat risk score (HRS) was an independent risk factor for predicting worse disease-free survival outcomes in the HRS-only risk model (hazard ratio = 3.274 [95% CI = 1.378–7.782]; *p* = 0.014) and combined habitat risk model (hazard ratio = 4.128 [95% CI = 1.744–9.769]; *p* = 0.003) in the validation cohort.

**Abstract:**

The purpose of this study was to identify perfusional subregions sharing similar kinetic characteristics from dynamic contrast-enhanced magnetic resonance imaging (MRI) using data-driven clustering, and to evaluate the effect of perfusional heterogeneity based on those subregions on patients’ survival outcomes in various risk models. From two hospitals, 308 and 147 women with invasive breast cancer who underwent preoperative MRI between October 2011 and July 2012 were retrospectively enrolled as development and validation cohorts, respectively. Using the Cox-least absolute shrinkage and selection operator model, a habitat risk score (HRS) was constructed from the radiomics features from the derived habitat map. An HRS-only, clinical, combined habitat, and two conventional radiomics risk models to predict patients’ disease-free survival (DFS) were built. Patients were classified into low-risk or high-risk groups using the median cutoff values of each risk score. Five habitats with distinct perfusion patterns were identified. An HRS was an independent risk factor for predicting worse DFS outcomes in the HRS-only risk model (hazard ratio = 3.274 [95% CI = 1.378–7.782]; *p* = 0.014) and combined habitat risk model (hazard ratio = 4.128 [95% CI = 1.744–9.769]; *p* = 0.003) in the validation cohort. In the validation cohort, the combined habitat risk model (hazard ratio = 4.128, *p* = 0.003, C-index = 0.760) showed the best performance among five different risk models. The quantification of perfusion heterogeneity is a potential approach for predicting prognosis and may facilitate personalized, tailored treatment strategies for breast cancer.

## 1. Introduction

Solid tumors are genomically, immunologically, and physiologically heterogeneous. These heterogeneous tumors are less likely to have durable responses to targeted and immune therapies [1,2,3] because treatment response is not uniform across the tumor, and therapy resistance occurs in different tumor regions. The complex vasculature within the tumor would lead to intratumoral heterogeneity. Tumors are supposed to have chaotic vasculature which leads to high permeability, and varying degrees of perfusion and oxygenation, and which has been proposed to be a major driver of the evolution of tumor heterogeneity at the genomic level and causes different microenvironment [4,5].

Since radiomics have been commonly used to measure intratumoral heterogeneity, radiomics analyses have conventionally been conducted for the whole tumor, and this approach assumes that the tumor is heterogeneous; however, the heterogeneity is evenly distributed throughout the tumor, thus neglecting regional phenotypic variations within a tumor [6]. Current tumor analyses using histograms [7] or radiomics analysis [8] focus on quantifying the heterogeneity and complexity by calculating the relationship between voxels [9]. In contrast to prior methodology, an emerging approach explicitly divides tumors into subregions containing clusters of voxels with similar characteristics, often called habitats, which allow better quantification of the intratumoral heterogeneity [10,11,12]. Habitat imaging is based on the speculation that identified subregions comprising voxels sharing similar imaging characteristics would share a common tumor biology [9,13]. Under this concept, Wu et al. [11] quantified intratumoral heterogeneity using dynamic contrast material-enhanced (DCE) magnetic resonance imaging (MRI) of breast cancers in neoadjuvant chemotherapy (NAC) settings to predict recurrence-free survival. They applied four metrics of DCE time-activity curves in each voxel and then used consensus clustering to divide the tumor into subregions.

Recently, Kim et al. [14] reported that higher values of kinetic heterogeneity, and peak enhancement determined using a commercially available computer-aided diagnosis (CAD) of preoperative MRI on the whole-tumor, were associated with worse recurrence outcomes in women with invasive breast cancer. Consequently, we hypothesized that intratumoral perfusion heterogeneity based on subregions of breast cancer derived from the kinetic features of DCE MRI maps would exhibit a better correlation with patient outcomes than that obtained from the conventional whole tumor. Therefore, the purpose of our study was to identify perfusional subregions sharing similar kinetic characteristics from DCE MRI using data-driven clustering, and to evaluate the effect of perfusional heterogeneity based on those subregions on patients’ survival outcomes in various risk models.

## 2. Materials and Methods

### 2.1. Patients

This retrospective multicenter study was conducted in accordance with the Declaration of Helsinki and was approved by the institutional review board of Samsung Medical Center (SMC 2017-08-136) and Gil Hospital (GDIRB 2016-088). The requirement for informed consent was waived due to the retrospective nature of the study and the analysis used anonymous clinical data.

Our cohort comprised breast cancer patients who had undergone surgery for invasive breast cancer from two hospitals between October 2011 and July 2012. Patients from Samsung Medical Center (SMC) were used as the development cohort, and patients from the Gil Hospital (GH) were used as the validation cohort. The flowchart in Figure 1 provides an overview of the datasets used in this study.

Our inclusion criteria were as follows: (a) preoperative DCE MRI, (b) initial unilateral breast malignancy with a final pathologic diagnosis of invasive breast cancer, and (c) lesion presenting as a mass on MRI. The exclusion criteria were as follows: (a) MRI performed in patients after the diagnosis of cancer by vacuum-assisted or excisional biopsy (*n* = 42); (b) MRI performed in patients treated with NAC (*n* = 92); (c) patients with a pre-existing malignancy in another organ (metastasis or primary malignancy) (*n* = 9); (d) involvement of any resection margin at final pathology (*n* = 16); (e) non-visualization of known breast cancer (*n* = 21); and (f) MRI quality inadequate for further processing (*n* = 14). Finally, 455 cancers in 455 women (mean age, 51 years; range, 24–85 years) were included. 

### 2.2. MRI Protocol

For the SMC cohort, all MRI scans were performed on a 1.5 T scanner from Philips (Achieva, Philips Healthcare, Best, The Netherlands). The MRI examination comprised turbo spin-echo T1- and T2-weighted sequences and a fat-suppressed 3-dimensional dynamic contrast-enhanced (DCE) sequence. Image subtraction was performed after the dynamic series. The DCE-MRI scans were acquired using the following parameters: TR/TE, 6.5/2.5; slice thickness, 3 mm; flip angle, 10°; matrix size, 376 × 374; and field of view, 32 × 32 cm. DCE-MRI was performed using axial imaging with one pre-contrast and six post-contrast dynamic series. After contrast injection, contrast-enhanced images were acquired at 0.5, 1.5, 2.5, 3.5, 4.5, and 5.5 min. A 0.1 mmol/kg bolus of gadobutrol (Gadovist; Bayer Healthcare Pharmaceutical, Berlin, Germany) was injected, followed by a 20 mL saline flush.

For the Gil cohort, all MRI scans were performed using a 3.0 T Philips scanner (Achieva, Philips Healthcare, Best, The Netherlands). The MRI examination consisted of turbo spin-echo T1- and T2-weighted sequences and a fat-suppressed 3-dimensional DCE sequence. Image subtraction was performed after the dynamic series. The DCE-MRI scans were acquired using the following parameters: TR/TE, 5.5/2.8; slice thickness, 3 mm; flip angle, 18°; matrix size, 424 × 424; and field of view, 34 × 34 cm. The DCE-MRI was performed using axial imaging, with one pre-contrast and five post-contrast dynamic series. Contrast-enhanced axial images were acquired at 1.5, 3, 4.5, and 6 min after contrast injection. A delayed sagittal image was obtained 8 min after the contrast injection. A 0.1 mmol/kg bolus of gadoterate meglumine (Dotarem; Guerbet, Villepinte, France and Clariscan; GE Healthcare, Oslo, Norway) was injected for dynamic contrast enhancement, followed by a 20 mL saline flush.

### 2.3. Clinicopathological Evaluation

The MRI findings were retrospectively evaluated according to the American College of Radiology Breast Imaging Reporting and Data System MR Lexicon [15] by two board-certified radiologists (S.Y.N. and E.S.K., with 12 and 15 years of experience in breast MRI, respectively). The radiologists assessed the shape (oval, round, irregular), margin (circumscribed, irregular, spiculated), and internal enhancement characteristics (homogeneous, heterogeneous, rim, dark internal septation) of each mass.

The final histopathological results of surgical specimens were reviewed to determine the following: pathologic diagnosis, histologic grade, presence of an extensive intraductal component (EIC), presence of lymphovascular invasion, estrogen receptor (ER), progesterone receptor (PR), human epidermal growth factor receptor 2 (HER2), p53, and Ki-67 expression status. Tumors with HER2 scores of 3+ (strong homogeneous staining) were considered positive. In the case of 2+ scores (moderate complete membranous staining in ≥1% of tumor cells), silver in situ hybridization was used to determine HER2 amplification. For convenience, the pathologic diagnoses were divided into three groups: invasive ductal carcinoma, invasive lobular carcinoma, and others.

The endpoint of our study was disease-free survival (DFS), which was defined as the time from the date of surgery to that of the first recurrence of the disease, of death, of the last-known evidence of the absence of disease, or of the most recent follow-up. Disease recurrence was defined as the outcome of breast cancer recurrence (local, regional, or distant) or new primary contralateral breast cancer (invasive or ductal carcinoma in situ). Patient medical records were used to obtain information regarding patient follow-up and recurrence status. Patients who did not have recurrence at the last follow-up or were lost to follow-up were treated as censored observations in the analyses. The last follow-up date was 1 September 2020.

### 2.4. Tumor Segmentation and Preprocessing

The pre-enhanced T1-weighted, early (1 min 30 s after contrast injection, respectively) and delayed (5 min 30 s for SMC, 6 min for GH after contrast injection, respectively) phases of contrast-enhanced T1-weighted MR images were retrieved from the Picture Archiving Communication System and loaded onto a workstation for further analysis. A region of interest (ROI) was manually drawn around the entire visible tumor on the early phase of contrast-enhanced T1-weighted images by a radiologist with 15 years of experience in breast MRI (E.S.K.) who was blinded to the clinical and pathological findings but was aware of the diagnosis of invasive carcinoma. The defined ROI was co-registered onto three other MRI series with a nine-parameter affine transform using mutual information as the similarity measure. The co-registration process allowed the researcher to define the ROI once and apply it to other imaging series of the same patient in a consistent manner. The ROI was drawn to be as large as possible but did not include edge voxels to avoid a partial volume effect. In the case of multifocal or multicentric disease, the largest tumor was selected as the index cancer for the analysis. Another radiologist (H.K.) drew another set of ROIs for a randomly selected 48 patients to assess interobserver reproducibility in terms of the intra-class correlation coefficient (ICC).

Each original imaging data were resampled to a 1 × 1 × 1 mm^3^ isotropic resolution using B-splice interpolation for accounting of the resolution differences in imaging resolutions. The ROIs defined in the original imaging data were resampled using nearest-neighbor interpolation on the isotropic resolution data. To harmonize the MRI intensity characteristics between the two cohorts, histogram-matching was applied to the validation cohort for conformance with the development cohort.

### 2.5. Generation of Perfusion Maps

Three perfusion parametric maps were constructed using pre-contrast, early, and delayed-phase images of the DCE MRI of the development cohort. The wash-in map ($E_{in}$) was generated by subtracting the pre-contrast image from the early phase image. The washout map ($E_{out}$) was generated by subtracting the delayed phase image from the early phase image. The washout ratio map ($R_{WO}$) was measured as the ratio between the signal intensity difference from the early phase to the delayed phase for the early phase. If the signal intensity of the delayed phase was increased compared to that of the early phase, it was considered as no washout. Each perfusion map was calculated on a voxel-by-voxel basis using the following equation:(1)$$E_{in}=I_{early}-I_{pre}$$
(2)$$E_{out}=I_{early}-I_{delayed}$$
(3)$$R_{WO}=\{\begin{array}{cc} \frac{I_{early}-I_{delayed}\;}{I_{early}}, & if\;I_{delayed}<I_{early}\\ 0, & otherwise \end{array}$$

### 2.6. Identifying Distinct Subregions Based on Perfusion Features with Population-Level Clustering 

A vector encompassing three perfusion features of each voxel (wash-in, washout, and washout ratio values) was defined as the perfusion feature vector (PFV). Each feature was quantized using a 256-bin histogram covering from the minimum to the maximum of each feature. For the development cohort, the PFVs of all patients were collected. We applied the k-means clustering algorithm with k values of 2 to 32 to identify a group of voxels with similar perfusion characteristics. Clustering was applied at the cohort level, not at the patient level, to ensure that clustering assignment remained consistent across patients. The Euclidean distance was used as the clustering cost measure. To select the optimal number of clusters (i.e., habitats), the clustering results were evaluated using the averaged Calinski-Harabasz score and Silhouette coefficient for each k value through 100 repetitions. The cluster centers from the development cohort were propagated to the validation cohort to ensure the application of the same clustering.

To investigate the characteristics of each subregion, we used box-and-whisker plots of three perfusion features and illustrated the kinetic profile of each subregion using the PFV centroid in a time–intensity curve. The portion of each subregion was also measured in relation to the total tumor volume.

### 2.7. Habitat Risk Score Building

To measure perfusion heterogeneity, we constructed a habitat map in which the intensity values of voxels were replaced with the previously identified habitat index. For example, if we identified five habitats, the habitat index varied from one to five. We adopted the concept of quantitative imaging features from radiomics studies to measure the spatial heterogeneity between the identified subregions. In total, 58 radiomics features were calculated using PyRadiomics from the habitat map. 

To evaluate the effect of perfusion heterogeneity measured from the habitat map using the radiomics technique for predicting patients’ survival outcomes, a habitat risk score (HRS) was constructed from the calculated features, which also included the proportions of each habitat. 

For this study, 4histograms, 24 gray-level co-occurrence matrix (GLCM), and 16 gray-level size zone matrix (GLSZM) features were calculated using PyRadiomics from the habitat map. We also computed the proportion of each habitat (i.e., the volume of each habitat divided by the tumor volume). Because the habitat map has only a few unique index values, the bin size was set to one for all feature calculation processes. Fourteen shape-based features from the entire tumor ROI were also computed using PyRadiomics. Each feature was z-score normalized based on the development cohort’s mean and standard deviation (SD). The L1-norm regularized Cox proportional hazard model (Cox-LASSO) was used to select features to build the HRS for DFS. The optimal coefficients were determined using a nested 10-fold cross-validation and grid search process. The HRS was defined as the relative risk at the initial time according to the following equation:(4)$$HRS_{i}=h\left(X_{i},0\right)=h_{0}\left(0\right)\cdot e^{\sum_{j=1}^{n}x_{ij}*\beta_{j}},$$
where $h\left(X_{i},0\right)$ denotes the initial hazard of the *i*th patient whose habitat feature vector is $X_{i}$, $x_{ij}$ denotes the *j*th prognostic habitat feature of the *i*th patient, *n* denotes the number of selected features, and $\beta_{j}$ denotes the corresponding Cox-LASSO coefficient of the *j*th habitat feature. The same HRS model was applied to the validation cohort, fixing the model parameters and using the feature values from the validation cohort to obtain the HRS for the validation cohort. We further conducted a Wilcoxon rank-sum test to compare the habitat risk score of the original set of ROIs with the additional set of ROIs which were randomly selected from 48 patients to assess interobserver reproducibility. 

### 2.8. Validation of HRS and Comparison of Performances among Different Risk Models

The Cox-least absolute shrinkage and selection operator (LASSO) model was used to analyze the effects of clinicopathological variables (age, histologic grade, pathologic type, T stage, N stage, EIC, lymphovascular invasion, ER, PR, HER2, p53, Ki-67, adjuvant chemotherapy, adjuvant radiation therapy, and adjuvant endocrine therapy); radiological variables (MRI findings of mass shape, mass margin, and internal enhancement pattern); and HRS on DFS. 

To demonstrate the value of the HRS, the HRS-only, clinical, and combined habitat risk models were constructed. To evaluate the additive effect of HRS for predicting survival outcome in the clinical risk model, the combined habitat risk model included 17 clinicopathological and radiological variables and HRS. To compare the predictive ability of the habitat-based method with the whole tumor-based method, we conducted two additional radiomics analyses on the DCE MRI of three phases and three perfusion maps generated from the DCE MRI. For these analyses, a total of 72 features were extracted. The radiomics risk score was also calculated using the Cox-LASSO model in the development cohort. 

The potential association of each risk score with DFS was first assessed in the development cohort and then validated in the validation cohort. Patients were classified into low-risk or high-risk groups using the median values of each risk score as cutoff values, which were also used for the validation cohort. The hazard ratio, *p*-value, and concordance index (C-index) were measured for all risk models, and we compared them among different models. 

### 2.9. Statistical Analysis

Patient characteristics in the development and validation cohorts were compared using a Student’s *t*-test for continuous variables and a chi-squared test or Fisher’s exact test for categorical variables. The characteristics of patients according to risk groups in the development cohort were also compared using Student’s t-test for continuous variables and chi-squared or Fisher’s exact tests for categorical variables. Different risk prediction models were compared with C-index values using the paired *t*-test [16]. All statistical analyses were conducted using the Statistics and Machine Learning Toolbox in MATLAB (R2019b).

## 3. Results

### 3.1. Patient Characteristics and Outcomes

Characteristics of the development and validation cohorts were compared, and the results are given in Table 1. 

Three-hundred and fifty-five (355/455, 78.0%) patients underwent breast-conserving surgery, and mastectomy was performed in 100 (100/455, 22.0%). There were forty-nine recurrences (twenty-three local-regional, nine contralateral breast, and seventeen distant recurrences) after a mean follow-up period of 84.1 months (range, 5–108 months). The mean time to recurrence was 39.3 months (range, 6–91 months). One patient had a recurrence within the first 6 months of follow-up, possibly due to residual disease.

### 3.2. Identified Subregions Based on Perfusion Features and Habitat Risk Score

The interobserver reliability in ROIs measured in intra-class correlation coefficient was on average 0.9237 over the habitat radiomics features. 

The optimal number of clusters was determined to be five based on the Calinski-Harabasz score and Silhouette coefficient (Figure 2). Resultantly, five subregions were determined in the development cohort, which were also applied in the validation cohort (Figure 3). Table 2 shows the cluster center and the proportion of each subregion among all voxels in the development cohort. Figure 4 shows the detailed distribution of the three perfusion features used (wash-in, washout, and washout ratio) in the development cohort and the characteristic illustration of each habitat using a time–intensity curve. In the early phase of contrast enhancement, habitats 2 and 3 showed strong wash-in, whereas habitats 1, 4, and 5 showed less wash-in. In the delayed phase of enhancement, habitat 1 showed a persistent pattern; habitats 2, 4, and 5 showed a washout pattern; and habitat 3 showed a plateau pattern. For each subregion, habitat 1 was the most predominant (41.9%), followed by habitats 3 (25.6%) and 4 (23.5%) (Table 2). The HRS was constructed using selected variables, mostly from texture features (Table 3). Notably, the proportion of habitats was not selected as a prognostic factor. 

The median HRS in the development cohort was 1.067 (range, 0.322–1.691; interquartile range, 0.823–1.255). Using this threshold value, the patients were classified into a high-risk (HRS ≥ 1.067) and a low-risk group (HRS < 1.067). The patient characteristics in the development cohort according to risk groups are shown in Table 4. In the development cohort, a higher T stage (*p* < 0.001), higher N stage (*p* < 0.001), higher histologic grade (*p* < 0.001), presence of lymphovascular invasion (*p* < 0.001), ER negativity (*p* < 0.001), PR negativity (*p* < 0.001), and HER2 positivity (*p* = 0.001) were associated with the high-risk group (Figure 5 and Figure 6). The HRS of the original set of ROIs and the additional set were comparable with a *p*-value of 0.7278.

### 3.3. Performance and Validation of the HRS 

Table 3 lists the features selected in various risk models using the Cox-LASSO model, where the selected features had non-zero coefficients. The coefficients represented the relative strength of the selected features and were reported in the third column. Higher N stage, presence of lymphovascular invasion, ER negativity, PR negativity, and Ki-67 ≥ 20% were selected for the clinical risk model as risk factors significantly associated with worse survival outcomes. The results of the combined habitat risk model showed that a higher HRS was associated with worse outcomes. The cutoff values of all risk models, including the two radiomics risk models, are presented in Table 5.

The risk stratification performance of each risk model is presented in Table 6. Higher radiomics risk scores calculated from two conventional radiomics risk models were independent risk factors for worse survival outcomes in the development cohort, but they were not reproducible in the validation cohort. The HRS was an independent risk factor for predicting worse outcomes in the HRS-only (hazard ratio = 3.274 [95% CI = 1.378–7.782]; *p* = 0.014) and combined habitat risk models (hazard ratio = 4.128 [95% CI = 1.744–9.769]; *p* = 0.003). When comparing the performance of the HRS-only risk model with that of the radiomics risk model based on perfusion maps, the C-index of the HRS-only risk model was 0.699, which was better than that of the radiomics_DCE_MR (C-index = 0.537) and radiomics_perfusion risk models (C-index = 0.640). The clinical risk model showed better performance than that of the HRS-only and the two radiomics risk models. When the HRS was combined with the clinical risk model, there was an improvement (C-index for combined habitat risk model = 0.760 vs. C-index for clinical risk model = 0.748, respectively), although it failed to show a statistical significance in both the development and validation cohorts (*p* = 0.342 and 0.456, respectively).

## 4. Discussion

Unlike previous radiomics studies on whole tumors, an emerging approach explicitly aimed at identifying distinct tumor areas or cell subpopulations is commonly called habitat imaging, which has been reported to reveal aggressive subregions that are important for determining prognosis and treatment response [6,17,18,19]. The use of complex signatures from multi-dimensional information and a radiomics analysis is a common framework for habitat imaging. Relative habitat volumes derived from clustering have been reported as predictors of survival [20,21]. Regarding histologic validation of the radiological habitat, some authors have reported detailed preclinical results through per-pixel spatial co-registration of the images and corresponding histologic findings of hypoxia, necrosis, and other conditions [22].

In this study, we focused on the spatial heterogeneity of kinetic profiles and hypothesized that intratumoral spatial heterogeneity might be reflected in the tumor enhancement kinetics of DCE MRI, and that this characteristic could be quantified using data-driven clustering analyses. As a relatively easy approach to measure perfusion heterogeneity, some authors have utilized CAD because it is popularly used and automatically provides quantitative values. They have revealed significant associations between peak enhancement or washout components as determined by CAD on preoperative MRI and recurrence-free survival in breast cancer patients [23,24]. Furthermore, a recent study conducted by Kim et al. [14] adopted the radiomics concept to evaluate the effect of kinetic patterns on survival outcomes. They found significant differences in early peak enhancement and delayed enhancement profiles, as determined by the CAD of preoperative breast MRIs, between the distant and non-distant metastasis groups. Importantly, they also noted a higher degree of kinetic heterogeneity in the distant metastasis group. However, they only used the kinetic characteristics of the delayed phase of enhancement (persistent, plateau, washout) and calculated the entropy measured from the amount of each delayed kinetic pattern to measure kinetic heterogeneity. As we believe that kinetic heterogeneity should include the characteristics of the early phase of enhancement as well, and consider not only the magnitude of each pattern but also their distribution, we proposed the HRS, which was defined as a measurement of heterogeneity in the habitat map that considered three perfusion features reflecting spatial heterogeneity among five distinct habitats. In our study, we identified five distinct intratumoral subregions combining kinetic profiles of both early and delayed phases of DCE MRI, and a higher HRS was an independent risk factor for predicting poor survival outcomes in the validation cohort. As far as we know, this study is the first to divide early and delayed phases by grouping them together. If the amount of washout is large, the prognosis is likely to be poor; however, it was not, and the most common type was type 1.

Interestingly, the proportion of each habitat did not affect the survival outcomes, which means that the distribution heterogeneity of habitats may be more important than the amount of each habitat. Our results could partially explain the previous results using CAD reporting inconsistent effects of kinetic patterns (i.e., washout component) with relatively low hazard ratios [14,23,24]. For example, in a study conducted by Kim et al. [23], a multivariate Cox analysis showed that a higher peak enhancement (hazard ratio = 1.001; *p* = 0.004) and a higher washout component (hazard ratio = 1.029; *p* = 0.017) were associated with poorer DFS. Conversely, in a study by Nam et al. [24], although the mean value of the washout component was higher in the recurrence group than in the non-recurrence group (39.19% vs. 38.08%, respectively), there was no significant difference in the DFS between the two groups (hazard ratio = 1.001; *p* = 0.834). A multivariate analysis revealed that a higher peak enhancement (hazard ratio = 1.004; *p* = 0.013) was independently associated with worse DFS outcomes. A more recent study by Kim et al. [14] reported that greater kinetic heterogeneity (hazard ratio = 19.2; *p* < 0.001) and higher peak enhancement (hazard ratio = 1.001; *p* = 0.045) were associated with worse distant metastasis-free survival in women with invasive breast cancer. Compared with these studies, our results showed that HRS was consistently an independent risk factor for poorer survival in the validation cohort (hazard ratio = 3.274; *p* = 0.014). We can confirm our hypothesis that spatial heterogeneity as well as magnitude of heterogeneity are important.

Several key aspects differentiate our study from previous studies. First and foremost, this was the first study to identify distinct patterns of enhancement profiles both in the early and delayed phases of enhancement, to quantify them from derived maps based on perfusional features, and to identify the relationship between perfusion heterogeneity and survival outcomes in the preoperative setting. Second, we applied well-grounded statistical principles to derive our habitat results. Third, we performed meticulous analyses to prove the better performance of the habitat-based method than that of the whole-tumor-based method. For this purpose, we conducted two additional radiomics analyses and compared the performances of the five different models. Consequently, radiomics analyses based on features calculated from the whole tumor (both on DCE MRI and perfusion maps) did not consistently enable the prediction of DFS in our study. Through these processes, our results provided evidence that a habitat-based analysis more robustly and consistently reflected the tumor characteristics than did the whole tumor-based analysis. One possible reason for the better performance of the habitat-based method could be the limitation of the voxel-based quantification of the whole-tumor approach, which can be easily affected by scan-related circumstances such as inhomogeneous fat suppression or blurring. However, habitat-based quantification lumps together similar voxels and is thus more robust in such circumstances. 

Our study had several limitations. First, although we called our five distinct subregions perfusion habitats, strict pathological correlations with image-based segmentations were lacking. However, such correlations are difficult to achieve. It is also close to a radiologic subgroup because the amount of each habitat did not affect the prognosis. Second, although the study results were validated in a separate cohort, only MRIs from the same vendor were utilized in this study. Technical factors such as field strength, repetition time, echo time, and flip angle might have influenced the results despite normalized imaging. Therefore, additional studies are required to confirm and validate our findings. However, 1.5T and 3T were mixed even though it was the same vendor. Due to the difference in scanners (1.5T in the development and 3T in the validation cohorts), the raw intensities could be different between scanners/cohorts. Thus, we mitigated this issue by matching the intensity histogram of the validation cohort to that of the development cohort for three phases (i.e., pre-contrast, early, and delay phases). This effectively normalized the intensity distribution of the validation cohort. Thirdly, patients’ characteristics between the two hospitals were very different. We speculate that it was probably due to the different nature of each hospital. However, our hypothesis was proven with statistical significance despite the inhomogeneous patient cohort. Finally, the technology used in this study seems very complex, making this technique seem impractical. To obtain greater clinical relevance, the development of easy-to-use software is warranted. However, our study provided a proof-of-concept. Similar to the developmental process of artificial intelligence for mammography, accumulating large-scale data using various machines or scanning parameters would make this technique practical. 

## 5. Conclusions

In conclusion, we quantified the spatial heterogeneity of the perfusional features of breast cancer on a derived habitat map from preoperative MRI scans. We identified five intratumoral subregions with distinct perfusion characteristics in breast cancer and showed that the spatial heterogeneity among the subregions was more important than the amount of each subregion. In addition to the clinical and pathologic factors, perfusion heterogeneity, defined by the spatial heterogeneity between perfusion habitats, was an independent predictor of DFS. The quantification of perfusion heterogeneity is a potential approach for predicting prognosis and can potentially lead to personalized, tailored treatment strategies for breast cancer.

## Figures and Tables

**Figure 1 cancers-14-01858-f001:**
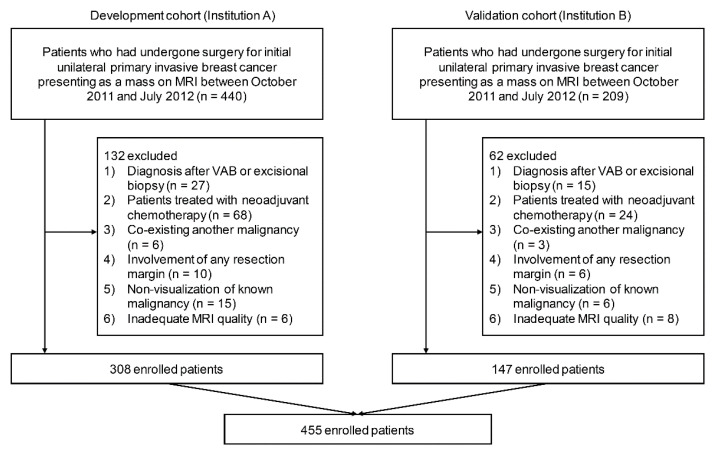
Flow chart of the study population. In total, 455 patients were included according to the inclusion and exclusion criteria from two hospitals.

**Figure 2 cancers-14-01858-f002:**
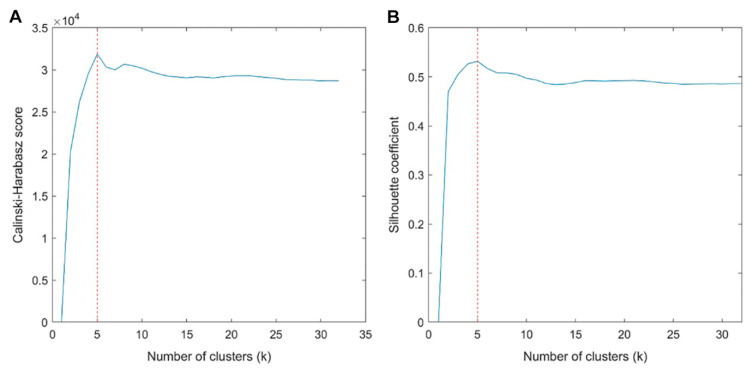
Calinski-Harabasz score plot (**A**) and Silhouette coefficient plots (**B**) to determine the optimal number of subregions (habitats). The values are averaged values from 100 repetitions. The red dotted line is the optimal value beyond which the scores started to decrease. The optimal number of clusters was determined as five.

**Figure 3 cancers-14-01858-f003:**
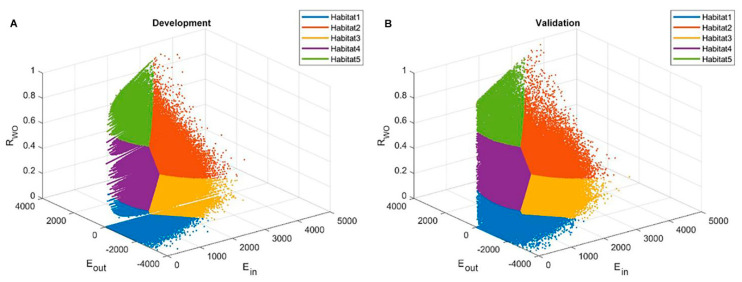
Habitat clustering. The five habitats defined by clustered voxel from normalized three perfusional maps are demonstrated. Clustering results of the development cohort are shown in (**A**), and the validation cohort is shown in (**B**). Clustering results were propagated into the validation cohort.

**Figure 4 cancers-14-01858-f004:**
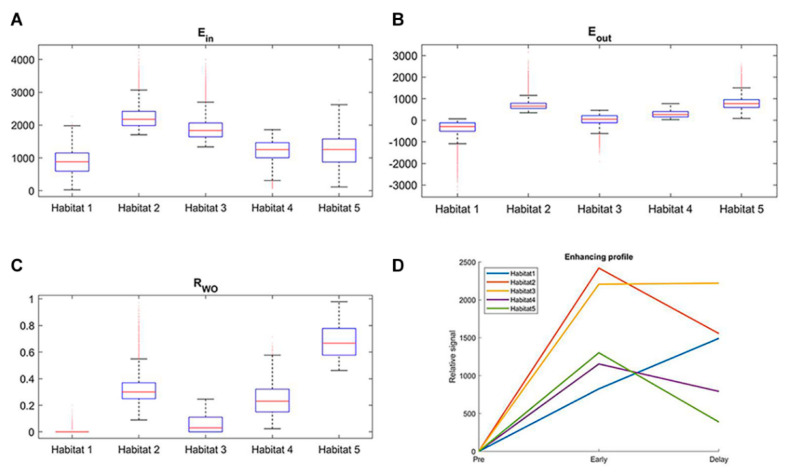
Box-and-whisker plots and illustrations of the kinetic profile of each habitat. (**A**–**C**) show the distribution of three perfusion features for five intratumoral subregions in the development cohort. (**D**) illustrates kinetic characteristics of each habitat with mean values using a time–intensity curve.

**Figure 5 cancers-14-01858-f005:**
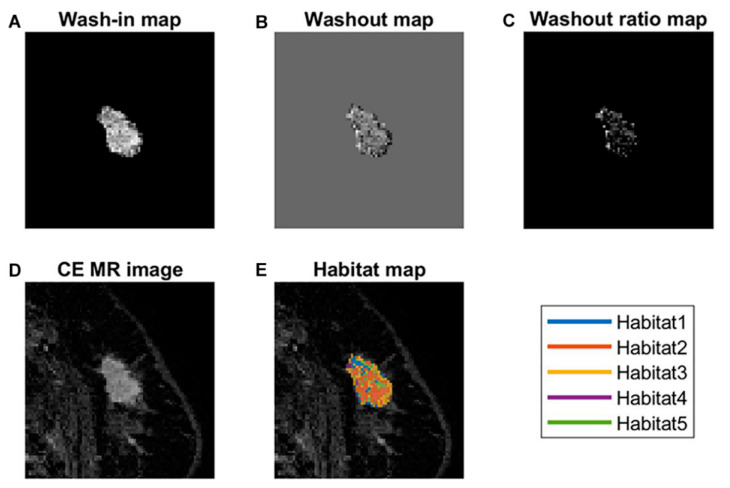
MRI of a 51-year-old woman with invasive ductal carcinoma in the left breast. (**A**) Wash-in map, (**B**) washout map, (**C**) washout ratio map, (**D**) early contrast-enhanced T1-weighted image, and (**E**) habitat map overlaid on the (**D**). In color overlay image (**E**), note marked heterogeneity of distribution of colors. In this patient, the habitat risk score was 1.258 and classified as a high-risk group. Nineteen months after surgery, a recurrence was detected in both lungs.

**Figure 6 cancers-14-01858-f006:**
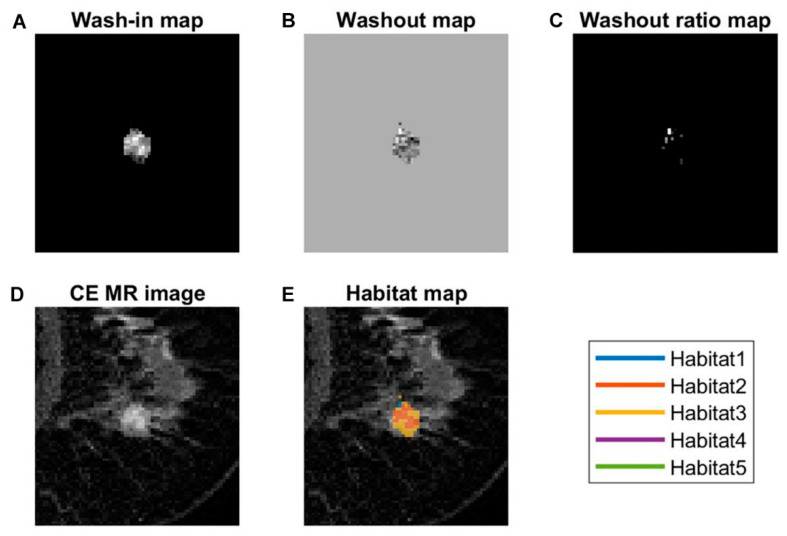
MRI of a 69-year-old woman with invasive ductal carcinoma in the right breast. (**A**) Wash-in map, (**B**) washout map, (**C**) washout ratio map, (**D**) early contrast-enhanced T1-weighted image, and (**E**) habitat map overlaid on the (**D**). In color overlay image (**E**), distribution heterogeneity of colors is not prominent compared to Figure 3. In this patient, the habitat risk score was 0.790 and classified as a low-risk group. During 85 months of follow-up, there was no evidence of recurrence.

**Table 1 cancers-14-01858-t001:** Characteristics of patients in the development and validation cohorts.

	Development Cohort (*n* = 308)	Validation Cohort (*n* = 147)	*p*-Value
**Age (y, means ± standard deviations)**	51.2 ± 10.5	49.1 ± 10.0	0.042
**T stage**			0.020
1	187 (60.7))	69 (46.9)	
2	112 (36.4)	73 (49.7)	
3	7 (2.3)	5 (3.4)	
4	2 (0.7)	0 (0.0)	
**N stage**			0.103
0	189 (60.4)	92 (62.6)	
1	98 (31.8)	37 (25.2)	
2	14 (4.6)	9 (6.1)	
3	7 (2.3)	9 (6.1)	
**Histologic grade**			<0.001
1	82 (26.6)	17 (11. 6)	
2	153 (49.7)	69 (46.9)	
3	73 (23.7)	66 (44.9)	
**Internal enhancement**			<0.001
Homogeneous	28 (9.1)	10 (6.8)	
Heterogeneous	203 (65.9)	129 (87.8)	
Rim enhancement	77 (25.0)	8 (5.4)	
**Mass Shape**			0.013
Round	24 (7.8)	7 (4.8)	
Oval	25 (8.1)	3 (2.0)	
Irregular	259 (84.1)	137 (93.2)	
**Mass Margin**			0.973
Circumscribed	17 (5.5)	8 (5.4)	
Not circumscribed	291 (94.5)	139 (94. 6)	
**Pathologic type**			0.280
IDC	285 (92.5)	133 (90.5)	
ILC	12 (3.9)	4 (2.7)	
Others	11 (3.6)	10 (6.8)	
**Lymphovascular invasion**			0.014
Positive	90 (29.2)	60 (40.8)	
Negative	218 (70.8)	87 (59.2)	
**Extensive intraductal component**			0.018
Positive	81 (26.3)	24 (16.3)	
Negative	227 (73.7)	123 (83.7)	
**ER**			0.001
Positive	244 (79.2)	95 (64.6)	
Negative	64 (20.8)	52 (35.4)	
**PR**			0.001
Positive	218 (70.8)	80 (54.4)	
Negative	90 (29.2)	67 (45.6)	
**p53**			<0.001
Positive	96 (31.2)	119 (81.0)	
Negative	212 (68.8)	28 (19.1)	
**HER2**			0.004
Positive	59 (19.2)	46 (31.3)	
Negative	249 (80.8)	101 (68.7)	
**Ki-67**			0.007
≥20%	144 (46.8)	91 (61.9)	
<20%	164 (53.3)	56 (38.1)	
**Adjuvant chemotherapy**			<0.001
No	110 (35.7)	28 (19.1)	
Yes	198 (64.3)	119 (81.0)	
**Adjuvant radiation therapy**			0.005
No	67 (21.8)	16 (10.9)	
Yes	241 (78.3)	131 (89.1)	
**Adjuvant endocrine therapy**			<0.001
No	63 (20.5)	52 (35.4)	
Yes	245 (79.6)	95 (64.6)	

Unless otherwise noted, data are numbers of patients and percentages are in parentheses.

**Table 2 cancers-14-01858-t002:** Cluster center and proportion of each subregion in the development cohort.

Habitat No.	Wash-In	Washout	Washout Ratio	Proportion
**Habitat 1**	825	−667	0.021	0.419
**Habitat 2**	2420	864	0.361	0.068
**Habitat 3**	2206	−14	0.075	0.256
**Habitat 4**	1154	362	0.338	0.235
**Habitat 5**	1302	915	0.717	0.022

**Table 3 cancers-14-01858-t003:** Various risk models and selected features in each model.

Category	Feature Name	Cox-LASSO Coefficient
**Habitat Risk Score**		
Shape	SVR	−0.235
GLCM	IDMN	−0.032
GLCM	IMC1	0.017
GLSZM	Small area emphasis	0.123
**Clinical risk model**		
Clinical	N stage	0.199
Clinical	Absence of lymphovascular invasion	−0.624
Clinical	ER negativity	0.398
Clinical	PR negativity	0.325
Clinical	Ki-67 < 20%	−0.623
**Combined habitat risk model**		
Habitat	Habitat Risk Score	1.451
Clinical	N stage	0.184
Clinical	Absence of lymphovascular invasion	−0.821
Clinical	ER negativity	0.416
Clinical	PR negativity	0.518
Clinical	p53	0.201
Clinical	Ki-67 < 20%	−0.551
Clinical	Adjuvant radiation therapy	−0.147

SVR, surface-to-volume ratio; IDMN, inverse difference normalized; IMC, informational measure of correlation; GLCM, gray-level co-occurrence matrix; GLSZM, gray-level size zone matrix.

**Table 4 cancers-14-01858-t004:** Clinicopathological characteristics according to risk groups based on HRS in the development cohort.

Characteristics	High-Risk (*n* = 154)	Low-Risk (*n* = 154)	*p*-Value
**Age (means ± standard deviations)**	50.8 ± 11.1	51.6 ± 9.9	0.494
**T stage**			<0.001
1	62 (40.3)	125 (81.2)	
2	88 (57.1)	24 (15.6)	
3	3 (2.0)	4 (2.6)	
4	1 (0.7)	1 (0.7)	
**N stage**			<0.001
0	66 (42.9)	123 (79.9)	
1	68 (44.2)	30 (19.5)	
2	13 (8.4)	1 (0.7)	
3	7 (4.6)	0 (0.0)	
**Histologic grade**			<0.001
1	13 (8.4)	69 (44.8)	
2	79 (51.3)	74 (48.1)	
3	62 (40.3)	11 (7.1)	
**Internal enhancement**			0.060
Homogeneous	11 (7.1)	17 (11.0)	
Heterogeneous	96 (62.3)	107 (69.5)	
Rim enhancement	47 (30.5)	30 (19.45)	
**Mass shape**			0.102
Round	10 (6.5)	14 (9.1)	
Oval	8 (5.2)	17 (11.0)	
Irregular	136 (88.3)	123 (79.9)	
**Mass margin**			0.088
Circumscribed	5 (3.3)	12 (7.8)	
Not circumscribed	149 (96.8)	142 (92.2)	
**Pathologic type**			0.308
IDC	145 (94.2)	140 (90.9)	
ILC	6 (3.9)	6 (3.9)	
Others	3 (2.0)	8 (5.2)	
**Lymphovascular invasion**			<0.001
Positive	75 (48.7)	15 (9.7)	
Negative	79 (51.3)	139 (90.3)	
**Extensive intraductal component**			0.897
Positive	40 (26.0)	41 (26.6)	
Negative	114 (74.0)	113 (73.4)	
**ER**			<0.001
Positive	94 (61.0)	150 (97.4)	
Negative	60 (39.0)	4 (2.6)	
**PR**			<0.001
Positive	79 (51.3)	139 (90.3)	
Negative	75 (48.7)	15 (9.7)	
**p53**			<0.001
Positive	63 (40.9)	33 (21.4)	
Negative	91 (59.1)	121 (78.6)	
**HER2**			0.001
Positive	41 (26.6)	18 (11.7)	
Negative	113 (73.4)	136 (88.3)	
**Ki-67**			<0.001
≥20%	119 (77.3)	25 (16.2)	
<20%	35 (22.7)	129 (83.8)	
**Adjuvant chemotherapy**			<0.001
No	23 (14.9)	87 (56.5)	
Yes	131 (85.1)	67 (43.5)	
**Adjuvant radiation therapy**			0.001
No	46 (29.9)	21 (13.6)	
Yes	108 (70.1)	133 (86.4)	
**Adjuvant endocrine therapy**			<0.001
No	57 (37.0)	6 (3.9)	
Yes	97 (63.0)	148 (96.1)	
**HRS** *	1.188 ± 0.208	0.896 ± 0.258	<0.001

Unless otherwise noted, data are numbers of patients and percentages are in parentheses. * Data are presented as mean ± standard deviation.

**Table 5 cancers-14-01858-t005:** Cutoff values for stratifying patients into low- and high-risk groups of each risk model.

Risk Model	Cutoff		Range	IQR
**Radiomics_DCE_MR**	1.118			
		Development	0.298–1.118	0.755–0.789
		Validation	0.325–1.794	0.875–0.927
**Radiomics_Perfusion**	1.043			
		Development	0.594–1.337	0.908–0.922
		Validation	0.611–1.327	0.892–0.915
**HRS-only**	1.067			
		Development	0.322–1.691	0.823–1.255
		Validation	0.396–1.533	0.872–1.181
**Clinical**	0.318			
		Development	0.170–1.816	0.170–0.654
		Validation	0.170–2.216	0.318–0.723
**Combined habitat**	1.867			
		Development	0.337–16.442	0.981–4.327
		Validation	0.321–22.717	1.132–5.186

**Table 6 cancers-14-01858-t006:** Comparison of performance between different risk prediction models.

	Development Cohort				Validation Cohort			
	Hazard Ratio (95% CI)	*p*-Value	C-Index (SE)	*p*-Value *	Hazard Ratio (95% CI)	*p*-Value	C-Index (SE)	*p*-Value *
**Radiomics_DCE_MR**	3.583 (1.708–7.516)	0.001	0.726 (0.002)	0.030	0.908 (0.385–2.146)	1.000	0.537 (0.005)	0.001
**Radiomics_Perfusion**	2.891 (1.377–6.065)	0.009	0.687 (0.002)	0.002	2.120 (0.866–5.191)	0.157	0.640 (0.005)	0.008
**HRS-only**	2.811 (1.340–5.898)	0.011	0.694 (0.003)	0.002	3.274 (1.378–7.782)	0.014	0.699 (0.005)	0.074
**Clinical**	4.523 (2.155–9.492)	<0.001	0.768 (0.002)	0.342	3.232 (1.330–7.855)	0.018	0.748 (0.004)	0.456
**Combined habitat**	5.227 (2.490–10.973)	<0.001	0.793 (0.002)	NA	4.128 (1.744–9.769)	0.003	0.760 (0.003)	NA

The *p*-value in each row refers to the *p*-value of the hazard ratio model. The *p*-value * is the model comparison between each model with the combined habitat model using the C-index values.

## Data Availability

The datasets used and analyzed in this study are not publicly available due to patient privacy requirements but are available upon reasonable request from the corresponding author.

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
