# Peer review of "Measurement of Perfusion Heterogeneity within Tumor Habitats on Magnetic Resonance Imaging and Its Association with Prognosis in Breast Cancer Patients"

_cancers, 2022, doi:10.3390/cancers14081858_

Round 1

Reviewer 1 Report

The manuscript entitled "Measurement of Perfusion Heterogeneity within Tumor Habitats on Magnetic Resonance Imaging and its Association with Prognosis in Breast Cancer Patients" provides a pretty comprehensive overview of effect of perfusional heterogeneity identified from DCE-MRI on overall survival. The manuscript is well written, thorough and only needs a few questions answered before publication.
1. Why did the authors use data from 2011-2012 ? Is there a specific reason for this ? 10 year old data can potentially have issues for the risk models due to outdated equipment, lower image quality than what is possible now in 2022, etc. Please provide a proper justification for this choice and the implications it would have on the analysis.

2. How does image quality affect the modeling ? Since the data used in the analysis are from different scanners (1.5T vs 3T) - this could potentially result in variability in the results. What are the author' thoughts on this ? 
This would be a good addition to the Discussion section.

3. The authors use ROIs drawn from only one radiologist which introduces biases. Ideally, for such studies, using multiple radiologists to cover inter and intra-observer variabilities is important. The authors should consider retrospectively repeating the analysis with atleast one other set of ROIs drawn by a different radiologist, to evaluate the effect on the final results. 

Reviewer 2 Report

This manuscript addresses breast tumor heterogeneity using a complex habitat risk score (HRS). The authors argue that HRS take into account the spatial heterogeneity of the tumor so it is more accurate than conventional radiomics approaches. To demonstrate the association between HRS and disease-free survival (DFS), a total of 455 patients from two hospitals were considered. Preoperative MRIs were analyzed and processed. Cox-LASSO based risk models were built from patient cohort from one hospital (development cohort) and models were applied to patient cohort from the second hospital (validation cohort). HRS showed statistically significant association with DFS in a combined risk model evaluated in the validation cohort.

In my opinion, this is a very complex manuscript with lots of details not clearly stated. It is difficult to understand. The concept of habitat imaging was new and there is no standard way of processing MR images. I have a few major concerns about this manuscript:

  1. It is important for this type of manuscript to clearly state methods. However, a lot of details were placed in supplementary. I suggest authors to move image processing and HRS calculation in Methods since both are important for readers to understand the methodology of this study.
  2. Development and validation cohorts need to be randomly selected from the whole cohort. As Table 1 shows, patient characteristics in these 2 cohorts were very different from each other.
  3. Kinetic parametric maps: wash-in and wash out were not normalized. This will be an issue for method generalization and cross-platform cross-vendor application.

In addition, I have a few minor revision suggestions:

  1. It was not clear whether tumors needed to have minimum size to be eligible in this study
  2. Please explain why patients received NAC were excluded
  3. Table 1 should be in Results
  4. Fonts in Figure 1 were too small
  5. It is not clear how features were selected in Table S2
  6. Various risk scores were used and cutoff points were used to separate patients into high-risk vs. low-risk groups. The methods should include how individual risk scores were constructed.
  7. Results were not well discussed in the Discussion.

Round 2

Reviewer 2 Report

The authors did substantial improvement on the manuscript. Fine to publish.